# Privacy without Noisy Gradients:
# Slicing Mechanism for Generative Model Training

**Kristjan Greenewald**
MIT-IBM Watson AI Lab, IBM Research
`kristjan.h.greenewald@ibm.com`

**Yuancheng Yu**
UIUC
`yyu51@illinois.edu`

**Hao Wang**
MIT-IBM Watson AI Lab, IBM Research
`hao@ibm.com`

**Kai Xu**
MIT-IBM Watson AI Lab, IBM Research
`xuk@ibm.com`

## Abstract

Training generative models with differential privacy (DP) typically involves injecting noise into gradient updates or adapting the discriminator's training procedure. As a result, such approaches often struggle with hyper-parameter tuning and convergence. We consider the *slicing privacy mechanism* that injects noise into random low-dimensional projections of the private data, and provide strong privacy guarantees for it. These noisy projections are used for training generative models. To enable optimizing generative models using this DP approach, we introduce the *smoothed-sliced $f$-divergence* and show it enjoys statistical consistency. Moreover, we present a kernel-based estimator for this divergence, circumventing the need for adversarial training. Extensive numerical experiments demonstrate that our approach can generate synthetic data of higher quality compared with baselines. Beyond performance improvement, our method, by sidestepping the need for noisy gradients, offers data scientists the flexibility to adjust generator architecture and hyper-parameters, run the optimization over any number of epochs, and even restart the optimization process—all without incurring additional privacy costs.

## 1 Introduction

Just as oil fueled the industrial revolution, data now propels innovation and progress in today's digital life. However, data sharing faces challenges due to privacy risks and regulatory policies (e.g., GDPR, CCPA, FTC). Synthetic data offers a promising solution as it closely resembles real data, retains its formats and essential properties, and can be seamlessly integrated into existing workflows. Nonetheless, modern generative models are vulnerable to privacy attacks that could expose individuals' sensitive information in the original data [HMDDC17, CYZF20, SOT22]. Today, differential privacy (DP) [DR14] stands as the de facto standard for privacy protection and it plays a crucial role in guiding the design of synthetic data. For example, DP was used in the release of microdata from Israel's National Registry of live births in 2014 to protect the privacy of mothers and newborns [HC24], as well as in the release of global victim-perpetrator data to ensure the privacy of victims [fM22]. Additionally, a recent publication [NDL+23] by the National Institute of Standards and Technology (NIST) recommends the use of DP algorithms in generating synthetic data to provide robust privacy protection against rapid developments in privacy attacks.

Existing approaches to training generative models often integrate DP by injecting noise into gradient updates or adapting the discriminator's training procedure [XLW+18, TKP19, CBV+21, TTCY21, BJWW+19, JYVDS19, TWB+19, BKZ23, NPA23, LJW+20, XGJ+22, COF20]. They provide several benefits, such as the ability to generate diverse data types (including continuous data, time-

series, and images), scalability to high-dimensional data, and accelerated runtime using GPUs. However, fine-tuning hyper-parameters within these frameworks can be challenging [NWD20, PS22, LT19]. Additionally, they encounter a dilemma when determining the number of training epochs: with a fixed privacy budget, increasing the number of epochs requires injecting more noise into gradient update per iteration, while fewer epochs might not be sufficient for the optimizer to converge. In such a setting, if the training does not converge, the only recourse may be to increase the privacy budget. This motivates a fundamental question:

*How can we train generative models with DP guarantees while ensuring easy fine-tuning, stable convergence, and high utility?*

In this paper, we introduce a new learning paradigm for training privacy-preserving generative models. Our approach decouples the training process into two steps: (i) computing noisy low-dimensional projections of the private data along random directions, and (ii) updating the generative models to fit these noisy projections. We establish DP guarantees for the first step and leverage the random projection to further tighten our privacy bound. This decoupling strategy offers several advantages. The post-processing property of DP ensures that any deep learning techniques can be applied in the second step. In other words, our approach is model-agnostic, allowing for smooth integration into existing training pipelines of generative models. With our method, data scientists have the flexibility to adjust generator architecture and hyper-parameters, optimize the generative model for any number of epochs, and even restart the optimization—all without worrying about additional privacy costs.

Based on this paradigm, we are motivated to introduce a new information-theoretic measure: the *smoothed-sliced $f$-divergence*. This divergence (randomly) projects the original and synthetic data distributions onto lower-dimensional spaces, followed by smoothing with isotropic Gaussian noise, and averaging their $f$-divergence over all projections. We prove that using this divergence as the loss function in generative model training is equivalent to the aforementioned two-step training process. Additionally, we present a kernel-based, differentiable estimator for this divergence. It circumvents the need for adversarial training in generative models, thereby enhancing convergence stability and robustness to different choices of hyper-parameters. Finally, we establish the statistical consistency of training generative models using this divergence.

In terms of the slicing mechanism, we build upon the work of [RL21]. They introduced the smoothed-sliced Wasserstein distance and applied it to generative models and domain adaptation tasks. However, their approach is limited to 1-dimensional projection spaces ($k = 1$), exploiting the closed-form expression of Wasserstein distances in 1D. Moreover, their privacy analysis contains a significant flaw in its derivation (see Remark 1 for detailed discussions). In contrast, we propose a generic framework for training privacy-preserving generative models, applicable to any $k$-dimensional projections. Empirically, we observe that setting the projection dimension to a small number (e.g., $k = 2, 3$) significantly improves the quality of synthetic data compared with $k = 1$ under the same privacy budget. Additionally, our kernel-based estimator eliminates the need for adversarial training in generative models, facilitating stable convergence. Finally, we present a completely new proof for establishing the DP guarantees of our framework. This analysis also applies to any smooth-sliced divergence objective, for instance, we provide corrected (and improved!) DP bounds for the smoothed-sliced Wasserstein framework of [RL21].

We demonstrate the effectiveness of our method through numerical experiments. We compare generative models trained using our method with those trained by standard privacy mechanisms (DP-SGD, PATE, or smoothed-sliced Wasserstein distance) among various real-world datasets. The results indicate that our approach consistently produces synthetic data of higher quality compared with baseline methods.

In summary, our main contributions are:

- We introduce a new framework for training privacy-preserving generative models. It offers easy hyper-parameter tuning and allows for optimizing generative models over any number of training epochs without extra privacy costs.

- We propose a new information-theoretic divergence and provide a kernel-based estimator. This estimator enables the training of generative models without relying on adversarial training, enhancing convergence stability.

- For the slicing mechanism, we extend the work of [RL21] and allow for projecting data onto any $k$-dimensional space. Using an entirely new proof technique, we provide DP guarantees, which in the $k = 1$ setting correct (and strengthen!) the privacy analysis in [RL21].

- We validate our method through numerical experiments. The results show that our method produces synthetic data of higher quality compared with baselines.

**Additional Related Work**

Recent work has proposed alternative approaches for training DP generative models without adversarial networks. For example, [CBV$^+$21] considered using the Sinkhorn divergence as the loss function, but their method adds noise to gradient updates to ensure DP, which leads to the challenges discussed in the introduction. Another line of work [HAP21, VCH$^+$22, HJSP23, YAS$^+$23] used the maximum mean discrepancy (MMD) to train DP generative models. They inject noise into the embedding of the private data distribution to maintain DP. However, minimizing their loss function to zero does not guarantee perfect matching between the synthetic and real data distributions, due to either not using a characteristic kernel or the approximation errors stemming from using finite-dimensional feature mappings to approximate the kernel function. In contrast, Proposition 1 proves that our smoothed-sliced $f$-divergence equals zero iff the synthetic and real distributions are identical; Corollary 1 establishes the statistical consistency of training generative models using our loss function. Additionally, we amplify our privacy bound by leveraging random projections in Theorem 1.

There is significant research introducing DP mechanisms tailored to generating tabular synthetic data [e.g., ZCP$^+$17, MSM19, MMS21, MMSM22, GAH$^+$14, VTB$^+$20, ABK$^+$21, ZWL$^+$21, LVW21, VAA$^+$22, DAHY24]. They select a set of workload queries (e.g., low-order marginal queries) and generates synthetic data to minimize approximation errors on these queries. These methods often maintain statistical properties of the original data with high accuracy, particularly for the selected workload queries; downstream predictive models trained on such synthetic data often achieve high performance when deployed on real data [TMH$^+$21, WSH$^+$23]. However, they only apply to categorical features[1], rely on special generative model architectures, or struggle to scale effectively to high-dimensional data. More broadly, there are several works analyzing DP synthetic data from a theoretical perspective or investigating other properties of synthetic data (e.g., missing values) [NTZ13, UV20, GMHI20, CLWX21, BSV22b, BSV22a, RTMT21, MZKH23].

A burgeoning line of work has explored slicing and smoothing to improve sample complexity in estimating divergence and optimal transport measures. These measures often suffer from extreme curses of dimensionality (e.g., $n^{-1/d}$ for Wasserstein distance [Dud69]). Previous studies have shown that both slicing [RPDB11, VFT$^+$19, LZC$^+$21, NDC$^+$20, GG21] and smoothing [GGNWP20, GG20] facilitate convergence at the parametric rate for both $f$-divergences and optimal transport distances, while preserving key properties of the original divergence (e.g., being zero iff the two distributions are identical). These modified divergences are also used as objective functions for training generative models. In our context, the use of smoothed-sliced divergence is less motivated by sample complexity (indeed using both smoothing and slicing would be unnecessary for achieving the parametric rate). Instead, it is motivated by the slicing privacy mechanism. Also, while we enjoy the sample complexity benefits of slicing, we do not benefit from the sample complexity improvements of smoothing [GGK20]. This is because achieving DP requires injecting a finite number (typically just one) of noise realizations per data point. Full smoothing results are still useful, however, for our consistency results in the asymptotic regime.

While the 1-dimensional sliced Wasserstein distance can be computed via a simple sorting algorithm [DZS18], there is in general no closed-form sample-based estimator for $f$-divergence, even in one dimension. While nearest neighbor [WKV09] and kernel-based [MSGHI18] estimators do exist, they often suffer from scalability issues and are not friendly to gradient-based optimization. Hence, it is a common practice to use dual forms of $f$-divergence in deriving adversarial-based training procedures for generative models [NCT16]. In the present work, we avoid this costly adversarial training using our moment matching estimator.

The supplementary material of this paper includes: (i) omitted proofs of all theoretical results and (ii) supporting experimental results.

---

[1]There are a few exceptions [e.g., LTVW23, VAA$^+$22] but they only generate a pre-determined number of synthetic samples, rather than providing a generative model, as ours does.

## 2 Preliminaries and Problem Setup

In this section, we review the concepts of differential privacy (DP), $f$-divergence, and a moment matching method used for estimating $f$-divergence.

### 2.1 Differential Privacy

We denote the original dataset as a matrix $\mathbf{X} \in \mathbb{R}^{n \times d}$, where $n$ denotes the number of records and $d$ represents the number of real-valued features per record.

**Definition 1** (Dataset adjacency). Two datasets $\mathbf{X}$ and $\mathbf{X}'$ are considered adjacent if they differ in a single row, say the $i$-th row, such that $\|\mathbf{X}_{i,:} - \mathbf{X}'_{i,:}\|_2 \leq 1$ where $\mathbf{X}_{i,:}$ and $\mathbf{X}'_{i,:}$ are the $i$-th row of $\mathbf{X}$ and $\mathbf{X}'$, respectively.

Next, we recall the definition of differential privacy (DP) [DR14].

**Definition 2** ($(\epsilon, \delta)$ differential privacy). A randomized mechanism $\mathcal{M} : \mathbb{R}^{n \times d} \to \mathbb{O}$ satisfies $(\epsilon, \delta)$-differential privacy if for any adjacent datasets $\mathbf{X}, \mathbf{X}'$ and all possible outcomes $\mathbb{S} \subseteq \mathbb{O}$, we have:
$$\Pr(\mathcal{M}(\mathbf{X}) \in \mathbb{S}) \leq \exp(\epsilon) \Pr(\mathcal{M}(\mathbf{X}') \in \mathbb{S}) + \delta.$$

DP has many compelling properties. The post-processing property states that if a mechanism $\mathcal{M}$ is $(\epsilon, \delta)$-DP, its outcome remains $(\epsilon, \delta)$-DP even after applying a (potentially randomized) function; the basic composition rule states that given a sequence of mechanisms $\mathcal{M}_1, \ldots \mathcal{M}_k$, if $\mathcal{M}_i$ is $(\epsilon_i, \delta_i)$-DP, then their composition $\mathcal{M}(\mathcal{D}) = (\mathcal{M}_1(\mathcal{D}), \ldots, \mathcal{M}_k(\mathcal{D}))$ will satisfy $(\sum_{i=1}^{k} \epsilon_i, \sum_{i=1}^{k} \delta_i)$-DP.

### 2.2 f-divergence and Moment Matching

We first recall the definition of $f$-divergence [Chapter 7 in PW23].

**Definition 3.** Let $f : (0, \infty) \to \mathbb{R}$ be a convex function with $f(1) = 0$ and $f(0) \triangleq f(0+)$. Let $P_X$ and $Q_X$ be two probability distributions on $\mathcal{X}$. If $P \ll Q$, then the $f$-divergence is defined as $\mathrm{D}_f(P\|Q) \triangleq \mathbb{E}_Q\left[f\left(\frac{\mathrm{d}P}{\mathrm{d}Q}\right)\right]$ where $\frac{\mathrm{d}P}{\mathrm{d}Q}$ is a Radon-Nikodym derivative. Additionally, we denote the density ratio by $r(x) \triangleq \frac{\mathrm{d}P}{\mathrm{d}Q}(x)$.

$f$-divergences have many nice properties. For example, it is always non-negative; and assuming $f$ is strictly convex at 1, then $\mathrm{D}_f(P\|Q) = 0$ if and only $P = Q$.

There is a burgeoning field of research focusing on $f$-divergence estimation [see e.g., GGNWP20, WY20, SXGS20]. Here we revisit a framework based on kernel mean matching, a special instance of moment matching methods [Chapter 3 in SSK12]. Given a reproducing kernel $\mathsf{K}(\boldsymbol{x}, \boldsymbol{x}')$, one can solve the following optimization problem to estimate the density-ratio function:
$$\min_{r \in \mathcal{R}} \left\| \int \mathsf{K}(\boldsymbol{x}, \cdot) \mathrm{d}P(\boldsymbol{x}) - \int \mathsf{K}(\boldsymbol{x}, \cdot) r(\boldsymbol{x}) \mathrm{d}Q(\boldsymbol{x}) \right\|_{\mathcal{R}}^2,$$
where $\|\cdot\|_{\mathcal{R}}$ denotes the norm of a reproducing kernel Hilbert space $\mathcal{R}$. One example of reproducing kernels is the Gaussian kernel $\mathsf{K}(\boldsymbol{x}, \boldsymbol{x}') = \exp\left(\frac{-\|\boldsymbol{x} - \boldsymbol{x}'\|_2^2}{2\sigma_g^2}\right)$. Given $\{\boldsymbol{x}_i^p\}_{i=1}^{n_p}$ drawn from $P$ and $\{\boldsymbol{x}_j^q\}_{j=1}^{n_q}$ drawn from $Q$, we can optimize an empirical version of the above optimization to obtain an analytical solution:
$$\hat{\boldsymbol{r}}_q = \frac{n_q}{n_p} \mathbf{K}_{q,q}^{-1} \mathbf{K}_{q,p} \mathbf{1}_{n_p},$$
where $\hat{\boldsymbol{r}}_q \in \mathbb{R}^{n_q}$ is the (empirically) optimal density ratio values at samples drawn from $Q$, $\mathbf{K}_{q,q} \in \mathbb{R}^{n_q \times n_q}$ and $\mathbf{K}_{q,p} \in \mathbb{R}^{n_q \times n_p}$ are the kernel Gram matrices:
$$[\hat{\boldsymbol{r}}_q]_j = \hat{r}\left(\boldsymbol{x}_j^q\right), \quad [\mathbf{K}_{q,q}]_{j,j'} = \mathsf{K}\left(\boldsymbol{x}_j^q, \boldsymbol{x}_{j'}^q\right), \quad [\mathbf{K}_{q,p}]_{j,i} = \mathsf{K}\left(\boldsymbol{x}_j^q, \boldsymbol{x}_i^p\right).$$

We extend the definition of $f$ to a vector by applying it to each element of the vector. Then we have
$$\hat{\mathrm{D}}_f(P\|Q) = \frac{1}{n_q} \mathbf{1}_{n_q}^T f(\hat{\boldsymbol{r}}_q) = \frac{1}{n_q} \mathbf{1}_{n_q}^T f\left(\frac{n_q}{n_p} \mathbf{K}_{q,q}^{-1} \mathbf{K}_{q,p} \mathbf{1}_{n_p}\right). \tag{1}$$

# 3 Main Results

We introduce a new information-theoretic measure—the smoothed-sliced $f$-divergence. We show that this divergence can measure the difference between distributions, with only access to noised $k$-dimensional slices of the distributions. This finding motivates a new DP mechanism: the $k$-*slicing privacy mechanism*. Finally, using the non-adversarial estimator (1), we apply the smoothed-sliced $f$-divergence as a new loss function to train privacy-preserving generative models.

## 3.1 Smoothed-sliced f-divergence

We start with giving a formal definition of the smoothed-sliced $f$-divergence.

**Definition 4.** Denote the Stiefel manifold of $d \times k$ matrices with orthonormal columns by $\mathbb{S}_k(\mathbb{R}^d)$. Let $\Theta \sim \mathrm{Unif}(\mathbb{S}_k(\mathbb{R}^d))$ and $\mathrm{N} \sim \mathcal{N}(\mathbf{0}, \sigma^2 \mathbf{I}_k)$. The smoothed-sliced $f$-divergence between distributions $P_{\mathrm{X}}$ and $Q_{\mathrm{X}}$ on $\mathbb{R}^d$ is defined as

$$
\begin{aligned}
\mathrm{SD}_{f,k,\sigma^2}(P_{\mathrm{X}} \| Q_{\mathrm{X}}) &\triangleq \mathrm{D}_f(P_{\Theta^T \mathrm{X} + \mathrm{N} | \Theta} \| Q_{\Theta^T \mathrm{X} + \mathrm{N} | \Theta} | P_\Theta) \\
&= \frac{1}{\mathrm{vol}(\mathbb{S}_k(\mathbb{R}^d))} \int_{\boldsymbol{\theta} \in \mathbb{S}_k(\mathbb{R}^d)} \mathrm{D}_f(P_{\boldsymbol{\theta}^T \mathrm{X} + \mathrm{N}} \| Q_{\boldsymbol{\theta}^T \mathrm{X} + \mathrm{N}}) \mathrm{d}\boldsymbol{\theta}.
\end{aligned}
$$

Next, we discuss some basic properties of this new divergence.

**Proposition 1.** *The smoothed-sliced $f$-divergence is non-negative:* $\mathrm{SD}_{f,k,\sigma^2}(P_{\mathrm{X}} \| Q_{\mathrm{X}}) \geq 0$ *for any* $k \geq 1$ *and* $\sigma \geq 0$. *If $f$ is strictly convex at $1$ and $P_{\mathrm{X}}, Q_{\mathrm{X}}$ have moment generating functions, then* $\mathrm{SD}_{f,k,\sigma^2}(P_{\mathrm{X}} \| Q_{\mathrm{X}}) = 0$ *if and only if* $P_{\mathrm{X}} = Q_{\mathrm{X}}$.

Given a set of real data $\{\boldsymbol{x}_i\}_{i=1}^n$ (i.e., rows of $\mathbf{X}$) and synthetic data $\{\boldsymbol{x}_i^{\mathrm{syn}}\}_{i=1}^{n_{\mathrm{syn}}}$, let $\hat{\mathrm{D}}_f$ denote any estimator of $k$-dimensional $f$-divergence (e.g., (1)). We draw random directions $\boldsymbol{\theta}_s$ and additive noise $\boldsymbol{v}_{s,i}$ and $\bar{\boldsymbol{v}}_{s,i}$. Then we can estimate the smoothed-sliced $f$-divergence by

$$
\widehat{\mathrm{SD}}_{f,k,\sigma^2}(P_{\mathrm{X}^{\mathrm{syn}}} \| P_{\mathrm{X}}) = \frac{1}{m} \sum_{s=1}^m \hat{\mathrm{D}}_f \left( \{\boldsymbol{x}_i^{\mathrm{syn}} \boldsymbol{\theta}_s + \bar{\boldsymbol{v}}_{s,i}\}_{i=1}^{n_{\mathrm{syn}}} \| \{\boldsymbol{x}_i \boldsymbol{\theta}_s + \boldsymbol{v}_{s,i}\}_{i=1}^n \right). \tag{2}
$$

As $\widehat{\mathrm{SD}}_{f,k,\sigma^2}$ is an empirical average of $m$ estimates of $k$-dimensional divergences, this estimator will inherit any statistical convergence bounds that apply to the chosen $\hat{\mathrm{D}}_f$.

## 3.2 Slicing Privacy Mechanism

The loss function in (2) accesses the original data solely through their noisy projections along random directions. This observation motivates the following $k$-slicing privacy mechanism.

**Definition 5.** Let $k$ denote the dimension of the random projections, and $m$ denote the number of them, yielding $m' = mk$. Let $\mathbf{X} \in \mathbb{R}^{n \times d}$ represent the original dataset where we assume $\|\mathbf{X}_{i,:}\|_2 \leq 1$ for all $i$. Let $\mathbf{U} \in \mathbb{R}^{d \times m'}$ and $\mathbf{V} \in \mathbb{R}^{n \times m'}$ be random slicing and noise matrices with each element independently drawn from $\mathrm{U}_{i,j} \sim \mathcal{N}(0, d^{-1})$ and $\mathrm{V}_{i,j} \sim \mathcal{N}(0, \sigma^2)$, respectively. The slicing privacy mechanism is defined as

$$
\mathcal{M}(\mathbf{X}) \triangleq (\mathbf{U}, \mathbf{X}\mathbf{U} + \mathbf{V}). \tag{3}
$$

**Remark 1.** Our mechanism outputs the random slicing matrix $\mathbf{U}$, as it will be used to project the synthetic data onto the same spaces during the training of generative models. In contrast, [RL21] did not include $\mathbf{U}$ in their privacy mechanism, leading them to give an incorrect derivation of the privacy guarantee. Similarly, our approach differs from using the Johnson-Lindenstrauss transform to preserve DP [BBDS12, EKKL20], as these methods do not reveal the random matrix.

In Definition 5, we draw random projections from a Gaussian distribution instead of uniformly from the Stiefel manifold. This choice simplifies the sampling procedure and streamlines our privacy analysis by making the the mechanism output jointly Gaussian with zero mean, conditioned on the data. Next, we establish the DP guarantees for our noisy slicing mechanism.

**Theorem 1.** *Assume $\gamma \triangleq \sigma^{-2}(\alpha^2 - \alpha) < d$. For all $\delta \in (0,1)$ and $\alpha > 1$, the mechanism $\mathcal{M}(\mathbf{X}) = (\mathbf{U}, \mathbf{X}\mathbf{U} + \mathbf{V})$ in Definition 5 satisfies*

$$\left( \frac{m'\alpha}{2\sigma^2(d-\gamma)} + \frac{\ln(1/\delta)}{\alpha - 1}, \delta \right) - DP.$$

*where $(\epsilon, \delta)$-DP is as defined in Definition 2 with dataset adjacency specified in Definition 1.*

Our privacy bound relies on $k$ (the dimension of projection spaces) and $m$ (the number of slices) only through their product $m' = mk$, which reveals a trade-off: with a fixed privacy budget, increasing $k$ aids generative models in capturing higher-order information of the original data, while increasing $m$ improves the accuracy of learning lower-order information. This trade-off resembles marginal-based mechanisms, where adjusting $k$ to enable generative models to learn the $k$-way marginal queries [Definition 1 in LVW21] involves a similar effect. However, we remark that even one-dimensional slices suffice to fully characterize a distribution since $\mathrm{SD}_{f,1,\sigma^2}(P_{\mathrm{X}} \| P_{\mathrm{X^{syn}}}) = 0$ implies $P_{\mathrm{X}} = P_{\mathrm{X^{syn}}}$ by Proposition 1. In contrast, matching all $k$-way marginal distributions for $k < d$ does not necessarily ensure $P_{\mathrm{X}} = P_{\mathrm{X^{syn}}}$.

We prove in Appendix Proposition 3 that the slicing privacy mechanism is $\left( \frac{m\alpha}{2\sigma^2} + \frac{\ln(1/\delta)}{\alpha - 1}, \delta \right)$-DP if the projection matrix $\mathbf{U}$ is deterministic. Comparing it with Theorem 1, we observe that by randomly selecting the projection matrix, we can achieve a tighter privacy bound by reducing a factor of $\frac{k}{d-\gamma}$ (for the first-term), even if $\mathbf{U}$ is disclosed by the privacy mechanism. The rationale behind this lies in the fact that a deterministic projection matrix allows the model designer/adversary to target specific individual records through carefully designed projection directions.

**Remark 2** (Choosing $\alpha$). The bound in Theorem 1 can be optimized in terms of $\alpha$ for a desired fixed $\delta$ and this optimization can be done numerically. Alternatively, there is an ad hoc strategy for 'approximately optimizing' $\alpha$. Let us assume $\gamma \leq d/2$, i.e. $\alpha^2 - \alpha \leq d\sigma^2/2$. Then

$$\epsilon \leq \frac{m'\alpha}{\sigma^2 d} + \frac{\ln(1/\delta)}{\alpha - 1}.$$

Minimizing the right hand side expression w.r.t. $\alpha > 1$ yields an optimal

$$\alpha^* = 1 + \sqrt{\frac{\sigma^2 d \ln(1/\delta)}{m'}}.$$

Substituting it into the (true) expression for $\epsilon$ yields

$$\epsilon^* = \frac{m'}{2\sigma^2(d - \gamma^*)} + \left( 1 + \sqrt{\frac{d}{2(d-\gamma)}} \right) \sqrt{\frac{m'\ln(1/\delta)}{\sigma^2 d}} \leq \frac{m'}{\sigma^2 d} + 2\sqrt{\frac{m'\ln(1/\delta)}{\sigma^2 d}},$$

where $\gamma^* = \sigma^{-2}(\alpha^2 - \alpha)$.

### 3.3 Training Generative Models

We outline our approach for training privacy-preserving generative models using the smoothed-sliced $f$-divergence (see Algorithm 1 for more details). First, we transform the output of the slicing privacy mechanism $\mathcal{M}(\mathbf{X}) = (\mathbf{U}, \mathbf{X}\mathbf{U} + \mathbf{V})$ into $\mathcal{O} = \{(\boldsymbol{\theta}_s, \boldsymbol{o}_{i,s})\}_{s \in [m], i \in [n]}$. Here $\boldsymbol{\theta}_s = \mathbf{U}_{:,(s-1)k+1:sk} \in \mathbb{R}^{d \times k}$ is the random slicing directions and $\boldsymbol{o}_{i,s} = (\mathbf{X}\mathbf{U} + \mathbf{V})_{i,(s-1)k+1:sk} \in \mathbb{R}^k$ is the (noisy) projection of the $i$-th data point onto the $s$-th slice.

Let $G_\omega$ be a generative model with trainable parameters $\omega \in \Omega$. At each iteration, we sample a mini-batch from $\mathcal{O}$ as $\{(\boldsymbol{\theta}_s, \boldsymbol{o}_{i,s})\}_{s \in [m], i \in [b]}$, where $b$ is the batch size.

To produce synthetic data, we draw random noise $\boldsymbol{z}_j$ for $j \in [b]$ and feed them into the generative model $\boldsymbol{x}_j^{\mathrm{syn}} = G_\omega(\boldsymbol{z}_j)$. Additionally, we draw random noise $\bar{\boldsymbol{v}}_{j,s} \sim \mathcal{N}(\mathbf{0}, \sigma^2 \mathbf{I}_k)$ for $j \in [b]$ and $s \in [m]$ that will be added to the projected synthetic data.[2] Given these samples, we train the generative model $G_\omega$ to generate samples close to the real data, using an estimator for the sliced

---

[2]This noise can be resampled every epoch as it does not touch the real data.

$f$-divergence to measure the discrepancy between the noisy-sliced real data $\boldsymbol{o}_{i,s}$ and the noisy-sliced synthetic data $\boldsymbol{x}_j^{\text{syn}}\boldsymbol{\theta}_s + \bar{\boldsymbol{v}}_{j,s}$. The smoothed slice $f$-divergence (2) yields the following loss function:

$$L(\omega) = \frac{1}{m} \sum_{s=1}^m \hat{\mathrm{D}}_f \left( \{ \boldsymbol{x}_j^{\text{syn}}\boldsymbol{\theta}_s + \bar{\boldsymbol{v}}_{s,j} \}_{j=1}^b \| \{ \boldsymbol{o}_{i,s} \}_{i=1}^b \right). \tag{4}$$

The following corollary is a direct application of Proposition 1.

**Corollary 1** (Consistency). *Consider a dataset* $\mathbf{X} \in \mathbb{R}^{n \times d}$ *of $n$ i.i.d. samples from a $d$-dimensional distribution $P_X$ whose moment generating function exists, and the slicing privacy mechanism $\mathcal{M}(\mathbf{X})$ as in Definition 5. Suppose the noise level $\sigma$ is fixed while $m \to \infty$ and $n \to \infty$.[3] Additionally, suppose that the chosen $f$-divergence estimator $\hat{\mathrm{D}}_f$ is consistent and $f$ is strictly convex at 1. Then $\lim_{n,m,b\to\infty} L(\omega^*) = 0$ for some $\omega^* \in \Omega$ if and only if $G_{\omega^*}(Z) \sim P_X$.*

Inserting our kernel-based estimator (1) yields for kernel K

$$L(\omega) = \frac{1}{mb} \sum_{s=1}^m \mathbf{1}_b^T f \left( \left( (\mathbf{K}_s + \tau \mathbf{I}_b)^{-1} \mathbf{K}_{s,\omega} \mathbf{1}_b \right)_+ \right), \tag{5}$$

where we compute the kernel Gram matrices $\mathbf{K}_s, \mathbf{K}_{s,\omega} \in \mathbb{R}^{b \times b}$ along each slice:

$$[\mathbf{K}_s]_{i,i'} = \mathsf{K}\left(\boldsymbol{o}_{i,s}, \boldsymbol{o}_{i',s}\right), \quad [\mathbf{K}_{s,\omega}]_{i,j} = \mathsf{K}\left(\boldsymbol{o}_{i,s}, \boldsymbol{x}_j^{\text{syn}}\boldsymbol{\theta}_s + \bar{\boldsymbol{v}}_{j,s}\right), \quad \text{for } s \in [m]. \tag{6}$$

To ensure the stability of computing matrix inversion, we include a $\tau \mathbf{I}_b$ where $\tau > 0$ is a small constant. Given that the domain of $f$ is $[0, \infty)$, we clip the density ratio estimator to make it non-negative.

**Remark 3.** Note that to compute the kernel Gram matrices $\mathbf{K}_s$ and $\mathbf{K}_{s,\omega}$, we need to apply the Gaussian kernel, which requires a constant $\sigma_g$. A heuristic of choosing this $\sigma_g$ is to make it the median distance between all samples—note that this will lead to different kernels along different slices. In practice, we use an ensemble of $\sigma_g$ and average their density ratio estimators.

---

**Algorithm 1** Training DP generative modes with the smoothed-sliced $f$-divergence.

---

**Input:** training data $\mathbf{X} = \{\boldsymbol{x}_i\}_{i=1}^n$; slicing dimension $k$; number of slices $m$; batch size; max number of iterations $T$; learning rate $\eta$; privacy budget $\epsilon, \delta$.
Apply the noisy slicing mechanism $\mathcal{M}(\mathbf{X})$ with $(\epsilon, \delta)$-DP, finding a noise variance $\sigma_{(\epsilon,\delta)}^2$ and transforming the output into $\{(\boldsymbol{\theta}_s, \boldsymbol{o}_{i,s})\}_{s\in[m],i\in[n]}$.
**for** $t = 1, \cdots, T$ **do**
    Sample a mini-batch of data from $\{(\boldsymbol{\theta}_s, \boldsymbol{o}_{i,s})\}_{s\in[m],i\in[n]}$, choosing a batch subset of the $i$.
    Generate synthetic data $\mathbf{X}_{\text{syn}}$ by feeding random noise into $G_\omega$.
    Slice and noise the synthetic data as $\boldsymbol{o}_s^{(\text{syn})} = \boldsymbol{\theta}_s \mathbf{X}_{\text{syn}} + \sigma_{(\epsilon,\delta)}\mathcal{N}(0,\mathbf{I})$, redrawing the noise each time (no privacy cost).
    Compute the kernel Gram matrices (6) under Gaussian kernels in each slice.
    Compute the loss function $L(\omega)$ in (5).
    Run stochastic-gradient optimization $\omega \leftarrow \omega - \eta \nabla_\omega L(\omega)$.
**end for**
**Output:** generative model $G_\omega$.

---

## 4 Numerical Experiments

We validate our approach and compare it with baselines through numerical experiments. Additional experimental results and details of our setup are reported in Appendix C.

### 4.1 Synthetic Tabular Data

**Baselines.** We compare our Algorithm 1 with four DP mechanisms: DP-SGD [XLW+18, RLP+20], PATE [JYVDS19], MERF [HAP21] and SliceWass [RL21]. Like our algorithm, these baselines can

---

[3]Privacy guarantees will be vacuous in this limit but this can be mitigated by privacy amplification (see Lemma 1 in appendix). Specifically, we can subsample $n$ data from a larger dataset of size $n'$ and let $n'/n \to \infty$.

| Dataset | DP Mechanism | Single Attribute Similarity | | Pairwise Attribute Similarity | | Classifier F1 Score |
|---|---|---|---|---|---|---|
| | | KSComp | TVComp | ContSim | CorrSim | LogitRegression |
| Income | Algorithm 1 | $0.40 \pm 0.05$ | $\mathbf{0.70} \pm 0.01$ | $\mathbf{0.35} \pm 0.01$ | $\mathbf{0.96} \pm 0.04$ | $\mathbf{0.31} \pm 0.20$ |
| | SliceWass | $0.24 \pm 0.05$ | $0.61 \pm 0.03$ | $0.27 \pm 0.02$ | $0.93 \pm 0.07$ | $0.13 \pm 0.14$ |
| | DP-SGD | $0.29 \pm 0.06$ | $0.38 \pm 0.02$ | $0.10 \pm 0.01$ | $0.75 \pm 0.07$ | $0.00 \pm 0.00$ |
| | PATE | $0.15 \pm 0.03$ | $0.40 \pm 0.05$ | $0.13 \pm 0.03$ | $0.90 \pm 0.07$ | $0.16 \pm 0.18$ |
| | MERF | $\mathbf{0.81} \pm 0.01$ | $0.51 \pm 0.04$ | $0.10 \pm 0.02$ | $0.56 \pm 0.03$ | $0.28 \pm 0.001$ |
| Coverage | Algorithm 1 | $\mathbf{0.74} \pm 0.07$ | $\mathbf{0.87} \pm 0.02$ | $\mathbf{0.63} \pm 0.02$ | $\mathbf{0.91} \pm 0.05$ | $0.41 \pm 0.04$ |
| | SliceWass | $0.72 \pm 0.06$ | $0.85 \pm 0.01$ | $0.60 \pm 0.01$ | $\mathbf{0.91} \pm 0.05$ | $\mathbf{0.41} \pm 0.02$ |
| | DP-SGD | $0.44 \pm 0.06$ | $0.63 \pm 0.09$ | $0.35 \pm 0.10$ | $0.73 \pm 0.15$ | $0.24 \pm 0.28$ |
| | PATE | $0.35 \pm 0.05$ | $0.51 \pm 0.03$ | $0.26 \pm 0.02$ | $0.85 \pm 0.09$ | $0.32 \pm 0.22$ |
| | MERF | $0.36 \pm 0.18$ | $0.52 \pm 0.04$ | $0.18 \pm 0.03$ | $0.69 \pm 0.15$ | $0.29 \pm 0.19$ |
| Mobility | Algorithm 1 | $0.68 \pm 0.06$ | $\mathbf{0.85} \pm 0.01$ | $\mathbf{0.50} \pm 0.01$ | $\mathbf{0.86} \pm 0.01$ | $\mathbf{0.69} \pm 0.03$ |
| | SliceWass | $\mathbf{0.74} \pm 0.04$ | $0.84 \pm 0.02$ | $0.50 \pm 0.02$ | $0.86 \pm 0.03$ | $0.67 \pm 0.06$ |
| | DP-SGD | $0.52 \pm 0.05$ | $0.70 \pm 0.06$ | $0.34 \pm 0.06$ | $0.74 \pm 0.13$ | $0.67 \pm 0.19$ |
| | PATE | $0.08 \pm 0.03$ | $0.54 \pm 0.03$ | $0.23 \pm 0.02$ | $0.83 \pm 0.02$ | $0.41 \pm 0.42$ |
| | MERF | $0.34 \pm 0.05$ | $0.59 \pm 0.01$ | $0.21 \pm 0.01$ | $0.68 \pm 0.04$ | $0.59 \pm 0.34$ |
| Employment | Algorithm 1 | $0.77 \pm 0.03$ | $\mathbf{0.83} \pm 0.01$ | $\mathbf{0.64} \pm 0.02$ | - | $0.47 \pm 0.07$ |
| | SliceWass | $0.74 \pm 0.04$ | $\mathbf{0.83} \pm 0.01$ | $0.62 \pm 0.00$ | - | $0.50 \pm 0.10$ |
| | DP-SGD | $0.45 \pm 0.04$ | $0.52 \pm 0.05$ | $0.25 \pm 0.05$ | - | $0.46 \pm 0.31$ |
| | PATE | $0.39 \pm 0.12$ | $0.54 \pm 0.04$ | $0.30 \pm 0.04$ | - | $0.38 \pm 0.28$ |
| | MERF | $\mathbf{0.96} \pm 0.01$ | $0.67 \pm 0.03$ | $0.34 \pm 0.02$ | - | $\mathbf{0.67} \pm 0.03$ |
| TravelTime | Algorithm 1 | $0.45 \pm 0.03$ | $\mathbf{0.75} \pm 0.01$ | $\mathbf{0.45} \pm 0.01$ | $0.87 \pm 0.05$ | $\mathbf{0.42} \pm 0.12$ |
| | SliceWass | $0.33 \pm 0.01$ | $0.62 \pm 0.02$ | $0.33 \pm 0.01$ | $0.84 \pm 0.05$ | $0.42 \pm 0.14$ |
| | DP-SGD | $0.37 \pm 0.08$ | $0.54 \pm 0.03$ | $0.22 \pm 0.03$ | $\mathbf{0.94} \pm 0.07$ | $0.37 \pm 0.33$ |
| | PATE | $0.37 \pm 0.04$ | $0.40 \pm 0.06$ | $0.15 \pm 0.03$ | $0.91 \pm 0.02$ | $0.34 \pm 0.20$ |
| | MERF | $\mathbf{0.61} \pm 0.04$ | $0.37 \pm 0.02$ | $0.07 \pm 0.01$ | $0.62 \pm 0.02$ | $0.14 \pm 0.28$ |

Table 1: We compare synthetic tabular data generated by our Algorithm 1 with baselines, all under the same privacy budget $\epsilon = 5.1$. We demonstrate them on the US Census data derived from the American Community Survey (ACS) with various evaluation metrics. These metrics range from 0 to 1, with higher scores indicating better performance. Since Employment has only one numerical column and CorrSim requires at least two numerical columns, we skip its values. The highest scores are highlighted in bold. If two methods have the same average score, only the one with lower standard deviation is highlighted. We remark that KSComp and CorrSim are less significant since they are designed for numerical columns. However, the benchmark datasets contain a limited number of numerical columns, with the majority being categorical (see Table 2).

handle both numerical and categorical data types without requiring data discretization and return a generative model that can produce any number of synthetic data. The implementations of the first two are adapted from an open-source Python library [Sma23], and the MERF implementation is from https://github.com/ParkLabML/DP-MERF. For SliceWass, we combine our slicing privacy mechanism with their main algorithm, which fixes the flaw in their privacy analysis and improves their privacy guarantees (and therefore quality results). Additionally, since their public Github repo does not include their synthetic data code, we implement their algorithm ourselves.

**Data.** We validate both our method and baselines using the US Census data derived from the American Community Survey (ACS) Public Use Microdata Sample (PUMS). Using the API of the Folktables package [DHMS21], we access the 2018 California data. Additionally, Folktables package provides five prediction tasks (Income , Coverage, Mobility, Employment, TravelTime) based

on a target column and a set of mixed-type features. Details about these data, including the number of records and columns, are provided in Table 2 in the appendix.

**Evaluation metrics.** We follow the evaluation principles in [TMH+21] for assessing the quality of synthetic data and leverage the APIs from an open-source library in our implementation [Dat23].

- `KSComplement` (numerical columns) and `TVComplement` (categorical columns). They measure the (average) similarity of one-way marginals (i.e., histograms of individual columns) between real and synthetic data.
- `ContingencySimilarity`. It measures the (average) similarity of pairs of categorical columns between real and synthetic data.
- `CorrelationSimilarity`. It measures the (average) correlations among numerical column pairs and computes the similarity between real and synthetic data.
- `BinaryLogisticRegression`. It measures the downstream classifier's F-1 score when trained on synthetic and test on real data.

Note that the benchmark datasets contain a limited number of numerical columns, with the majority being categorical (see Table 2). Hence, the applicability of `CorrelationSimilarity` and `KSComplement` is constrained as they are tailored for numerical columns.

**Main results and observations.** We present the experimental results for $\epsilon = 5.1$ in Table 1 and show results for $\epsilon = 8.1$ in Appendix C. As shown, the two methods using the slicing privacy mechanism (Algorithm 1 and `SliceWass`) consistently outperform the other baselines. `MERF` has better numbers for three instances of `KSComp` and one of `LogitRegression`, but is signficantly worse in other instances and for the remaining metrics. For both Algorithm 1 and `SliceWass`, we used the same hyper-parameter initialization and neural network architecture across all datasets. We believe that with more extensive hyperparameter tuning, which incurs no extra privacy cost as previously discussed, their performance could be even further improved.

Algorithm 1 exhibits a more favorable performance compared with `SliceWass` in most settings. The rationale behind this observation lies in the fact that `SliceWass` is limited to 1-dimensional slices ($k = 1$), relying on the closed-form expression of Wasserstein distances in 1D. In contrast, our Algorithm 1 is applicable to higher-dimensional slices. This capability enables the generative models to capture higher-order statistical information, leading to better performance in the pairwise `ContingencySimilarity` statistic and higher robustness in the `BinaryLogisticRegression` downstream task.

## 4.2 Synthetic Image Data

We conduct an experiment to generate DP synthetic image data using the MNIST dataset [LBBH98]. We train separate generative models for each of the 10 classes in MNIST, with 10% of the data randomly sampled for each class. We evaluate the quality of the synthetic images by measuring the downstream accuracy of a classifier trained on the synthetic data and deployed on the real MNIST test set. We compare Algorithm 1 and `SliceWass` (combined with our slicing privacy mechanism), with `MERF`, which is a state-of-the-art MMD-based method for generating synthetic images [HAP21]. We observe that `MERF` outperforms the two slicing-mechanism-based algorithms at lower privacy budgets but underperforms at higher budgets. We hypothesize that `MERF` achieves superior performance in the low-budget regime since it privatizes only the mean embedding of the data for each class; its generative model is designed to recover this mean rather than the full data distribution. As using only the mean involves very little privacy budget, they are able to add less noise and outperform in the low-budget regime. Our approach, which models the full data distribution, does not

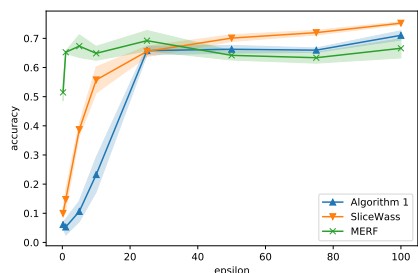

Figure 1: We compare accuracy for downstream classification as a function of privacy budget ($\epsilon$) for synthetic MNIST data created by `MERF` with our Algorithm 1 and `SliceWass`. Note that the two slicing-mechanism-based approaches outperform `MERF` for higher privacy budgets.

benefit from these savings. However, as the privacy budget increases, our method better captures the true data distribution, leading to improved results over `MERF`.

### 4.3 Domain Adaptation

Following the experiments in [RL21], for completeness we also apply our smoothed-sliced $f$-divergence in domain adaptation tasks. We benchmark our approach with `SliceWass` [RL21] using the implementation available on their Github repo, modified to use our DP bound. This experiment aims to privately train a classifier using labeled data from a source domain and unlabelled data from a target domain. We consider the target and source domains being MNIST and USPS, as well as the reverse. Results for varying $\epsilon$ are depicted in Figure 2. As shown, our Algorithm 1 using 100 slices improves on `SliceWass`.

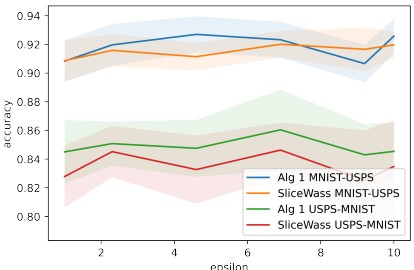

Figure 2: Unsupervised domain adaptation between from MNIST to USPS and vice versa.

## 5 Final Remarks and Limitations

In this paper, we consider the slicing mechanism for training privacy-preserving generative models and derive strong privacy guarantees for it. This mechanism motivates us to combine it with $f$-divergence to yield a smoothed-sliced $f$-divergence that can be estimated with a kernel-based density ratio estimator. Our approach circumvents the need for injecting noise into gradient updates and avoids adversarial training for optimizing generative models. It provides flexibility in selecting neural architectures and tuning hyper-parameters without incurring additional privacy costs. Through experiments on synthetic data generation, we validate our approach and compare it with existing baselines. Our findings suggest that the slicing privacy mechanism is a powerful tool for training generative models to create private synthetic data, and the smoothed-sliced $f$-divergence provides a promising avenue for advancing the field of privacy-preserving data synthesis in sensitive, high-dimensional datasets. Additionally, we hope our effort can be of particular interest to the information theory community and open up a new application frontier for classical information-theoretic tools.

There are several promising avenues worth exploring. The $f$-divergence has many nice properties, including (strong) data processing inequalities, inequalities among different $f$-divergences, and variational representation. It would be interesting to investigate whether similar properties extend to the smoothed-sliced $f$-divergence $\mathrm{SD}_{f,k,\sigma^2}$. Additionally, deriving sample-complexity bounds for estimating $\mathrm{SD}_{f,k,\sigma^2}$ from finite samples would offer valuable insights. Finally, the use of synthetic data presents both opportunities and challenges. On the one hand, synthetic data is well-suited for various tasks like early model development, educational demonstrations, simulation, and testing. On the other, synthetic data can never fully replicate all aspects of the original data, and adding DP guarantees introduces an additional layer of complexity requiring a careful balance between privacy and utility. Therefore, any machine learning models trained on synthetic data or any insights drawn from synthetic data should undergo thorough evaluation before deployment in real-world scenarios. If biases are detected, it is crucial to diagnose their source—whether stemming from the synthetic data generation algorithm, the noise added to the real data, or other factors—to ensure that the models perform robustly and reliably in practice.

### Acknowledgement

We would like to thank Akash Srivastava for his insights on connecting sliced divergences to synthetic data generation, and for sharing [RL21] with us.

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

# A  Privacy Analysis

Before diving into the proof of Theorem 1, we will first revisit the definition of Rényi differential privacy (RDP) [Mir17, DR16, BS16] and its connection with differential privacy (DP). Additionally, we will recall a useful property of DP known as privacy amplification by subsampling.

**Definition 6.** A randomized mechanism $\mathcal{M} : \mathbb{R}^{n \times d} \to \mathbb{O}$ satisfies $(\alpha, \epsilon)$-RDP if for any adjacent datasets $\mathbf{X}, \mathbf{X}' \in \mathbb{R}^{n \times d}$, we have

$$\sup_{\mathbf{X} \sim \mathbf{X}'} D_\alpha(\mathcal{M}(\mathbf{X}) \| \mathcal{M}(\mathbf{X}')) \leq \epsilon,$$

where $D_\alpha$ is the Rényi-$\alpha$ divergence.

**Proposition 2** (Prop. 3 in [Mir17])**.** *If $\mathcal{M}$ is an $(\alpha, \epsilon)$-RDP mechanism, it also satisfies $(\epsilon + \frac{\ln(1/\delta)}{\alpha - 1}, \delta)$-DP for all $\delta \in (0, 1)$.*

We remind readers that two datasets $\mathbf{X}$ and $\mathbf{X}'$ are considered adjacent if they differ in a single row, say the $i$-th row, such that $\|\mathbf{X}_{i,:} - \mathbf{X}'_{i,:}\|_2 \leq 1$ where $\mathbf{X}_{i,:}$ and $\mathbf{X}'_{i,:}$ are the $i$-th row of $\mathbf{X}$ and $\mathbf{X}'$, respectively. This condition applies, for example, to unbounded DP (add/remove one record) when the $L_2$ norm of each record is upper bounded by 1.

We revisit privacy amplification by subsampling, a useful technique for handling large-scale datasets that can save computational resources and memory. It proposes that the privacy guarantees of a DP mechanism can be improved by randomly subsampling the private dataset before applying the DP mechanism [KLN+11, LQS12, BBG18].

**Lemma 1.** *Given a dataset $\mathbf{X} = [\boldsymbol{x}_1^T, \cdots, \boldsymbol{x}_n^T]^T$, PoissonSample$_\tau$ independently draws Bernoulli random variables $\sigma_i \sim \mathsf{Bern}(\tau)$ for $i \in [n]$ and outputs a subset $[\boldsymbol{x}_i^T]_{\sigma_i=1, i \in [n]}^T$. If a mechanism $\mathcal{M}$ is $(\epsilon, \delta)$-DP, then $\mathcal{M} \circ$ PoissonSample$_\tau$ satisfies $(\epsilon', \delta')$-DP where $\epsilon' = \log(1 + \tau(e^\epsilon - 1))$ and $\delta' = \tau\delta$.*

We introduce some notations and a result from [GAL13] that will be used in proving the following lemma. For a matrix $\mathbf{A} \in \mathbb{R}^{m \times n}$, its determinant is denoted by $|\mathbf{A}|$. The vectorization of $\mathbf{A}$ is defined as

$$\text{vec}(\mathbf{A}) \triangleq [a_{1,1}, \cdots, a_{m,1}, \cdots, a_{1,n}, \cdots, a_{m,n}]^T.$$

For two matrices $\mathbf{A} \in \mathbb{R}^{m \times n}$ and $\mathbf{B} \in \mathbb{R}^{p \times q}$, their Kronecker product $\mathbf{A} \otimes \mathbf{B}$ is the $pm \times qn$ block matrix:

$$\mathbf{A} \otimes \mathbf{B} \triangleq \begin{bmatrix} a_{1,1}\mathbf{B} & \cdots & a_{1,n}\mathbf{B} \\ \vdots & \ddots & \vdots \\ a_{m,1}\mathbf{B} & \cdots & a_{m,n}\mathbf{B} \end{bmatrix}.$$

We recall a result from [GAL13, Table 2]:

$$D_\alpha(\mathcal{N}(\boldsymbol{\mu}, \boldsymbol{\Sigma}) \| \mathcal{N}(\boldsymbol{\mu}', \boldsymbol{\Sigma}'))$$
$$= \frac{\alpha}{2}(\boldsymbol{\mu} - \boldsymbol{\mu}')^T((1-\alpha)\boldsymbol{\Sigma} + \alpha\boldsymbol{\Sigma}')^{-1}(\boldsymbol{\mu} - \boldsymbol{\mu}') - \frac{1}{2(\alpha-1)} \ln \frac{|(1-\alpha)\boldsymbol{\Sigma} + \alpha\boldsymbol{\Sigma}'|}{|\boldsymbol{\Sigma}|^{1-\alpha}|\boldsymbol{\Sigma}'|^\alpha}, \quad (7)$$

whenever $\alpha\boldsymbol{\Sigma}^{-1} + (1-\alpha)\boldsymbol{\Sigma}'^{-1} > 0$.

Next, we establish the RDP guarantees for our mechanism.

**Lemma 2.** *Assume $\alpha > 1$ and $\gamma \triangleq \sigma^{-2}(\alpha^2 - \alpha) < d$. Let $\mathbf{U} \in \mathbb{R}^{d \times m'}$ and $\mathbf{V} \in \mathbb{R}^{n \times m'}$ be two random matrices whose elements are drawn independently from $\mathcal{N}(0, d^{-1})$ and $\mathcal{N}(0, \sigma^2)$, respectively. Then the mechanism $\mathcal{M}(\mathbf{X}) = (\mathbf{U}, \mathbf{X}\mathbf{U} + \mathbf{V})$ satisfies $(\alpha, \epsilon)$-RDP where*

$$\epsilon = \frac{m'\alpha}{2\sigma^2(d - \gamma)}.$$

*Proof of Lemma 2.* Since $\mathbf{X}$ is not random, the mechanism $\mathcal{M}(\mathbf{X})$ maps $\mathbf{X}$ to a $(dm' + nm')$-variate Gaussian distribution with mean zero and covariance

$$
\begin{aligned}
\text{Cov}\left(\begin{bmatrix} \text{vec}(\mathbf{U}^T) \\ \text{vec}((\mathbf{X}\mathbf{U} + \mathbf{V})^T) \end{bmatrix}\right) &= d^{-1} \begin{bmatrix} \mathbf{I}_{dm'} & \mathbf{X}^T \otimes \mathbf{I}_{m'} \\ \mathbf{X} \otimes \mathbf{I}_{m'} & (\mathbf{X}\mathbf{X}^T \otimes \mathbf{I}_{m'}) + d\sigma^2 \mathbf{I}_{nm'} \end{bmatrix} \\
&= d^{-1} \begin{bmatrix} \mathbf{I}_d & \mathbf{X}^T \\ \mathbf{X} & \mathbf{X}\mathbf{X}^T + d\sigma^2 \mathbf{I}_n \end{bmatrix} \otimes \mathbf{I}_{m'} \\
&= d^{-1} \mathbf{B} \otimes \mathbf{I}_{m'},
\end{aligned}
\tag{8}
$$

where we define $\mathbf{B} \triangleq \begin{bmatrix} \mathbf{I}_d & \mathbf{X}^T \\ \mathbf{X} & \mathbf{X}\mathbf{X}^T + d\sigma^2 \mathbf{I}_n \end{bmatrix}$. Combining (7) with (8) yields

$$
\begin{aligned}
\text{D}_\alpha(\mathcal{M}(\mathbf{X}) \| \mathcal{M}(\mathbf{X}')) &= -\frac{m'}{2(\alpha - 1)} \ln \frac{|(1-\alpha)\mathbf{B} + \alpha\mathbf{B}'|}{|\mathbf{B}|^{1-\alpha}|\mathbf{B}'|^\alpha} \\
&= \frac{m'}{2(\alpha - 1)} \left[ ((1-\alpha)\ln|\mathbf{B}| + \alpha \ln|\mathbf{B}'|) - \ln|(1-\alpha)\mathbf{B} + \alpha\mathbf{B}'| \right].
\end{aligned}
$$

Next, we compute the determinants in the above Rényi divergence. Using the formula for determinants of block matrices, we know

$$
|\mathbf{B}| = |\mathbf{B}'| = |d\sigma^2 \mathbf{I}_n|.
$$

For the linear combination, we have

$$
\begin{aligned}
&|(1-\alpha)\mathbf{B} + \alpha\mathbf{B}'| \\
&= \left| \begin{bmatrix} \mathbf{I}_d & (1-\alpha)X^T + \alpha\mathbf{X}'^T \\ (1-\alpha)\mathbf{X} + \alpha\mathbf{X}' & (1-\alpha)\mathbf{X}\mathbf{X}^T + \alpha\mathbf{X}'\mathbf{X}'^T + d\sigma^2 \mathbf{I}_n \end{bmatrix} \right| \\
&= \left| d\sigma^2 \mathbf{I}_n + \left[ (1-\alpha)\mathbf{X}\mathbf{X}^T + \alpha\mathbf{X}'\mathbf{X}'^T - ((1-\alpha)\mathbf{X} + \alpha\mathbf{X}')((1-\alpha)\mathbf{X} + \alpha\mathbf{X}')^T \right] \right|.
\end{aligned}
$$

We denote

$$
\begin{aligned}
\mathbf{\Delta} &\triangleq (1-\alpha)\mathbf{X}\mathbf{X}^T + \alpha\mathbf{X}'\mathbf{X}'^T - ((1-\alpha)\mathbf{X} + \alpha\mathbf{X}')((1-\alpha)\mathbf{X} + \alpha\mathbf{X}')^T \\
&= -\alpha(\mathbf{X}\mathbf{X}^T - \mathbf{X}'\mathbf{X}'^T) - \alpha^2(\mathbf{X} - \mathbf{X}')(\mathbf{X} - \mathbf{X}')^T + \alpha\mathbf{X}(\mathbf{X} - \mathbf{X}')^T + \alpha(\mathbf{X} - \mathbf{X}')\mathbf{X}^T \\
&= -\alpha(-(\mathbf{X} - \mathbf{X}')(\mathbf{X} - \mathbf{X}')^T + \mathbf{X}(\mathbf{X} - \mathbf{X}')^T + (\mathbf{X} - \mathbf{X}')\mathbf{X}^T) \\
&\quad - \alpha^2(\mathbf{X} - \mathbf{X}')(\mathbf{X} - \mathbf{X}')^T + \alpha\mathbf{X}(\mathbf{X} - \mathbf{X}')^T + \alpha(\mathbf{X} - \mathbf{X}')\mathbf{X}^T \\
&= (\alpha - \alpha^2)(\mathbf{X} - \mathbf{X}')(\mathbf{X} - \mathbf{X}')^T.
\end{aligned}
$$

Since $\mathbf{X}$ and $\mathbf{X}'$ only differ in the $i$-th row, only one entry of this matrix can be nonzero, specifically,

$$
\alpha - \alpha^2 \le \Delta_{ii} \le 0.
$$

Then

$$
\begin{aligned}
\text{D}_\alpha(\mathcal{M}(\mathbf{X}) \| \mathcal{M}(\mathbf{X}')) &= \frac{m'}{2(\alpha - 1)} \left[ ((1-\alpha)\ln|\mathbf{B}| + \alpha \ln|\mathbf{B}'|) - \ln|(1-\alpha)\mathbf{B} + \alpha\mathbf{B}'| \right] \\
&= \frac{m'}{2(\alpha - 1)} \left[ \ln|d\sigma^2 \mathbf{I}_n| - \ln|d\sigma^2 \mathbf{I}_n + \mathbf{\Delta}| \right] \\
&= \frac{m'}{2(\alpha - 1)} \left[ \ln(d\sigma^2) - \ln(d\sigma^2 + \Delta_{ii}) \right].
\end{aligned}
$$

Recall the assumption that $|\Delta_{ii}| \leq \alpha^2 - \alpha \leq \gamma\sigma^2$ for $\gamma \in (0, d)$. Then using the fact that $\log(a)$ is concave in $a$ and has derivative $1/a$ for $a > 0$,

$$
\begin{aligned}
D_\alpha(\mathcal{M}(\mathbf{X}) \| \mathcal{M}(\mathbf{X}')) &\leq -\frac{m'}{2(\alpha-1)} \frac{1}{d\sigma^2 + \Delta_{ii}} \Delta_{ii} \\
&\leq \frac{m'}{2(\alpha-1)} \frac{1}{d\sigma^2 + (\alpha - \alpha^2)} (\alpha^2 - \alpha) \\
&\leq \frac{m'}{2(\alpha-1)} \frac{1}{(d-\gamma)\sigma^2} (\alpha^2 - \alpha) \\
&\leq \frac{m'}{2(d-\gamma)\sigma^2(\alpha-1)} (\alpha^2 - \alpha) \\
&\leq \frac{m'\alpha}{2\sigma^2(d-\gamma)}.
\end{aligned}
$$

$\square$

*Proof of Theorem 1.* Theorem 1 follows directly from combining Lemma 2 with Proposition 2. $\square$

Next, we establish a privacy bound assuming the projection matrix $\mathbf{U}$ is deterministic. Comparing this with Theorem 1, we observe that by randomly selecting the projection matrix, we can achieve a tighter privacy bound, even if it is disclosed by the privacy mechanism.

**Proposition 3.** *Let* $\mathbf{X} \in \mathbb{R}^{n \times d}$ *represents the original dataset. Let* $\mathbf{V} \in \mathbb{R}^{n \times m'}$ *be a random noise matrix with each element independently drawn from* $\mathbf{V}_{i,j} \sim \mathcal{N}(0, \sigma^2)$. *Let* $\mathbf{U} \in \mathbb{R}^{d \times m'}$ *be a deterministic matrix such that for any* $j \in [m]$

$$\mathbf{U}_{:,(j-1)k+1}, \cdots, \mathbf{U}_{:,jk} \text{ are orthonormal}$$

*The privacy mechanism* $\mathcal{M}_{det}(\mathbf{X}) \triangleq \mathbf{XU} + \mathbf{V}$ *satisfies*

$$\left( \frac{m\alpha}{2\sigma^2} + \frac{\ln(1/\delta)}{\alpha - 1}, \delta \right) - DP.$$

*Proof of Proposition 3.* The privacy mechanism $\mathcal{M}_{\text{det}}$ maps $\mathbf{X}$ to a $nm'$-variate Gaussian distribution with mean and covariance being:

$$\text{mean}(\mathbf{XU} + \mathbf{V}) = \text{vec}(\mathbf{XU}), \quad \text{Cov}(\mathbf{XU} + \mathbf{V}) = \sigma^2 \mathbf{I}_{nm'}.$$

Using (7) yields

$$
\begin{aligned}
D_\alpha(\mathcal{M}(\mathbf{X}) \| \mathcal{M}(\mathbf{X}')) &= \frac{\alpha}{2} \text{vec}(\mathbf{XU} - \mathbf{X}'\mathbf{U})^T (\sigma^2 \mathbf{I}_{nm'})^{-1} \text{vec}(\mathbf{XU} - \mathbf{X}'\mathbf{U}) \\
&= \frac{\alpha}{2\sigma^2} \text{vec}((\mathbf{X} - \mathbf{X}')\mathbf{U})^T \text{vec}((\mathbf{X} - \mathbf{X}')\mathbf{U}) \\
&= \frac{\alpha}{2\sigma^2} \|(\mathbf{X}_{i,:} - \mathbf{X}'_{i,:})\mathbf{U}\|_2^2.
\end{aligned}
$$

Without loss of generality, we assume

$$\mathbf{X}_{i,:} - \mathbf{X}'_{i,:} = [x, 0, \cdots, 0] \quad \text{with } |x| \leq 1.$$

Additionally, recall the assumption that for any $j \in [m]$, $\mathbf{U}_{:,(j-1)k+1}, \cdots, \mathbf{U}_{:,jk}$ are orthonormal. Hence, $U_{1,(j-1)k+1}^2 + \cdots + U_{1,jk+1}^2 \leq 1$, leading to

$$D_\alpha(\mathcal{M}(\mathbf{X}) \| \mathcal{M}(\mathbf{X}')) \leq \frac{\alpha}{2\sigma^2} \sum_{j=1}^{m'} U_{1,j}^2 \leq \frac{\alpha m}{2\sigma^2}.$$

As a result, our privacy mechanism is $(\alpha, \frac{\alpha m}{2\sigma^2})$-RDP. Combining it with Proposition 2 yields the desired DP guarantee. $\square$

# B    Other Omitted Proofs

*Proof of Proposition 1.* The property $\mathrm{SD}_{f,k,\sigma^2}(P_X\|Q_X) \geq 0$ follows by the non-negativity of the $f$-divergence in the integrand. If $P_X = Q_X$, then clearly

$$\mathrm{SD}_{f,k,\sigma^2}(P_X\|Q_X) = \frac{1}{\mathrm{vol}(\mathbb{S}_k(\mathbb{R}^d))} \int_{\boldsymbol{\theta} \in \mathbb{S}_k(\mathbb{R}^d)} \mathrm{D}_f(P_{\boldsymbol{\theta}^T X + N}\|Q_{\boldsymbol{\theta}^T X + N}) \mathrm{d}\boldsymbol{\theta} = 0$$

since the $f$-divergences $\mathrm{D}_f(P_{\boldsymbol{\theta}^T X + N}\|Q_{\boldsymbol{\theta}^T X + N})$ are all zero by definition.

Conversely, by the nonnegativity of the $f$-divergence, $\mathrm{SD}_{f,k,\sigma^2}(P_X\|Q_X) = 0$ implies

$$\mathrm{D}_f(P_{\boldsymbol{\theta}^T X + N}\|Q_{\boldsymbol{\theta}^T X + N}) = 0$$

for almost every $\boldsymbol{\theta} \in \mathbb{S}_k(\mathbb{R}^d)$. Additionally, since $f$ is strictly convex at 1, we have

$$P_{\boldsymbol{\theta}^T X + N} = Q_{\boldsymbol{\theta}^T X + N}.$$

As a result, by the invertibility of Gaussian convolution,

$$P_{\boldsymbol{\theta}^T X} = Q_{\boldsymbol{\theta}^T X}$$

for almost every $\boldsymbol{\theta} \in \mathbb{S}_k(\mathbb{R}^d)$ since $f$-divergence nullifies iff the arguments are equal. Since the distributions are equal, the moment generating functions of these projections are equal, i.e. for all $t \in \mathbb{R}^k$

$$\mathbb{E}_{Z \sim P_{\boldsymbol{\theta}^T X}}[e^{t^T Z}] = \mathbb{E}_{Z \sim Q_{\boldsymbol{\theta}^T X}}[e^{t^T Z}]$$

i.e.

$$\mathbb{E}_{X \sim P_X}[e^{t^T \boldsymbol{\theta}^T X}] = \mathbb{E}_{X \sim Q_X}[e^{t^T \boldsymbol{\theta}^T X}].$$

Since $\boldsymbol{\theta} t$ is dense in $\mathbb{R}^d$ when $\boldsymbol{\theta} \in \mathbb{S}_k(\mathbb{R}^d)$ and $t \in \mathbb{R}^k$, we have that for any $s \in \mathbb{R}^d$,

$$\mathbb{E}_{X \sim P_X}[e^{s^T X}] = \mathbb{E}_{X \sim Q_X}[e^{s^T X}].$$

But these are just the moment generating functions of $P_X$ and $Q_X$. Since the moment generating functions exist and are equal for all $s$, the densities $P_X$, $Q_X$ must be equal.

$\square$

*Proof of Corollary 1.* By the assumption that $\hat{\mathrm{D}}_f$ is a consistent estimator of the $f$-divergence, $b \to \infty$ with $\sigma$ constant implies that for any fixed $\omega$,

$$\lim_{n,b \to \infty} L(\omega) = \frac{1}{m} \sum_{s=1}^{m} \mathrm{D}_f\left(P_{\boldsymbol{\theta}_s^T G_\omega(Z) + N}\|P_{\boldsymbol{\theta}_s^T X + N}\right), \tag{9}$$

where $N \sim \mathcal{N}(\mathbf{0}, \sigma^2 \mathbf{I}_k)$.

In the first case, we assume $G_{\omega^*}(Z) \sim P_X$. Substituting $\omega = \omega^*$ into (9) yields

$$\lim_{n,b \to \infty} L(\omega^*) = \frac{1}{m} \sum_{s=1}^{m} \mathrm{D}_f\left(P_{\boldsymbol{\theta}_s^T X + N}\|P_{\boldsymbol{\theta}_s^T X + N}\right) = 0,$$

where equality with zero follows because the $f$-divergence nullifies if its arguments are the same distribution.

For the second case, we assume that for some $\omega^* \in \Omega$

$$\lim_{m,n,b \to \infty} L(\omega^*) = \lim_{m \to \infty} \frac{1}{m} \sum_{s=1}^{m} \mathrm{D}_f\left(P_{\boldsymbol{\theta}_s^T G_{\omega^*}(Z) + N}\|P_{\boldsymbol{\theta}_s^T X + N}\right) = 0.$$

Since the $\mathrm{D}_f\left(P_{\boldsymbol{\theta}_s^T G_{\omega^*}(Z) + N}\|P_{\boldsymbol{\theta}_s^T X + N}\right)$ terms are all $\geq 0$, the fact that this limit exists implies that

$$\lim_{m,n,b \to \infty} L(\omega^*) = \frac{1}{\mathrm{vol}(\mathbb{S}_k(\mathbb{R}^d))} \int_{\boldsymbol{\theta} \in \mathbb{S}_k(\mathbb{R}^d)} \mathrm{D}_f(P_{\boldsymbol{\theta}^T G_{\omega^*}(Z) + N}\|P_{\boldsymbol{\theta}^T G_{\omega^*}(Z) + N}) \mathrm{d}\boldsymbol{\theta}$$

$$= \mathrm{SD}_{f,k,\sigma^2}(P_{G_{\omega^*}(Z)}\|P_X).$$

By Proposition 1, this can equal zero only if $P_{G_{\omega^*}(Z)} = P_X$.

$\square$

| Dataset | #Records | #Columns | #Categorical Cols | #Numerical Cols |
|---|---|---|---|---|
| `Income` | 195,665 | 10 | 8 | 2 |
| `Coverage` | 138,554 | 19 | 17 | 2 |
| `Mobility` | 80,329 | 21 | 19 | 2 |
| `Employment` | 378,817 | 16 | 15 | 1 |
| `TravelTime` | 172,508 | 16 | 14 | 2 |

Table 2: Dataset details. In addition to the described columns, each dataset has a binary outcome column, which is used by the evaluation metric `LogitRegression`.

## C  Details on the Experimental Results

**Datasets**  We demonstrate our method and baselines using the US Census data derived from the American Community Survey (ACS) Public Use Microdata Sample (PUMS). Using the API of the Folktables package [DHMS21], we access the 2018 California data. Additionally, Folktables package provides five prediction tasks (`Income` , `Coverage`, `Mobility`, `Employment`, `TravelTime`) based on a target column and a set of mixed-type features. We provide more details about these datasets in Table 2.

**Data pre-processing.**  We pre-process each dataset using the open-source Python library [Sma23]. It normalizes numerical columns and one-hot encodes categorical columns, all within a privacy-preserving way using an $\epsilon = 0.5$. Note that the pre-processing budget is not included in the stated $\epsilon$ in our results. To satisfy the bounded norm assumption in our privacy analysis, we normalize the resulting dataset so that the largest row 2-norm is upper bounded by 1. This can be achieved by e.g., normalizing each row by $1/\sqrt{d}$, where $d$ is the number of columns. We subsample each dataset at a rate of $0.25$ using the Poisson mechanism, which amplifies the DP guarantees according to Lemma 1. For all our experiments, we report the amplified $\epsilon$.

**More details.**  For our method and `SliceWass`, all experiments used batch size of 128 and learning rate $2 \cdot 10^{-5}$, and ran for 200 epochs. For our method and baselines, each model was trained using a V100 GPU, with runtimes typically less than 2 hours for our method (200 epochs).

**Additional results.**  We repeat our experiments with a different privacy budget $\epsilon = 8.1$ and report the results in Table 3.

| Dataset | DP Mechanism | Single Attribute Similarity | | Pairwise Attribute Similarity | | Classifier F1 Score |
| | | KSComp | TVComp | ContSim | CorrSim | LogitRegression |
|---|---|---|---|---|---|---|
| Income | Algorithm 1 | 0.40 ±0.03 | **0.73** ±0.01 | **0.37** ±0.01 | 0.94 ±0.01 | 0.35 ±0.17 |
| | SliceWass | 0.25 ±0.01 | 0.61 ±0.01 | 0.27 ±0.00 | **0.98** ±0.01 | 0.17 ±0.03 |
| | DP-SGD | 0.30 ±0.10 | 0.37 ±0.02 | 0.09 ±0.02 | 0.91 ±0.06 | 0.15 ±0.30 |
| | PATE | 0.19 ±0.09 | 0.40 ±0.05 | 0.12 ±0.03 | 0.95 ±0.02 | **0.38** ±0.21 |
| | MERF | **0.82** ±0.01 | 0.47 ±0.03 | 0.08 ±0.01 | 0.60 ±0.14 | 0.30 ±0.12 |
| Coverage | Algorithm 1 | **0.72** ±0.02 | **0.87** ±0.01 | **0.63** ±0.01 | **0.88** ±0.06 | 0.41 ±0.07 |
| | SliceWass | 0.71 ±0.04 | 0.86 ±0.01 | 0.62 ±0.01 | 0.87 ±0.03 | 0.45 ±0.10 |
| | DP-SGD | 0.46 ±0.02 | 0.71 ±0.03 | 0.42 ±0.05 | 0.52 ±0.11 | 0.27 ±0.19 |
| | PATE | 0.32 ±0.03 | 0.51 ±0.05 | 0.25 ±0.05 | 0.83 ±0.03 | **0.50** ±0.02 |
| | MERF | 0.44 ±0.15 | 0.52 ±0.01 | 0.18 ±0.01 | 0.61 ±0.003 | 0.48 ±0.10 |
| Mobility | Algorithm 1 | 0.71 ±0.02 | **0.85** ±0.01 | **0.50** ±0.01 | **0.86** ±0.01 | **0.70** ±0.01 |
| | SliceWass | **0.76** ±0.04 | 0.85 ±0.02 | 0.50 ±0.02 | 0.86 ±0.04 | 0.65 ±0.05 |
| | DP-SGD | 0.52 ±0.06 | 0.72 ±0.06 | 0.35 ±0.06 | 0.72 ±0.15 | 0.62 ±0.20 |
| | PATE | 0.08 ±0.04 | 0.53 ±0.03 | 0.22 ±0.02 | 0.83 ±0.01 | 0.54 ±0.40 |
| | MERF | 0.33 ±0.09 | 0.58 ±0.01 | 0.20 ±0.01 | 0.68 ±0.03 | 0.19 ±0.08 |
| Employment | Algorithm 1 | 0.67 ±0.00 | **0.83** ±0.00 | **0.63** ±0.00 | - | 0.52 ±0.00 |
| | SliceWass | 0.75 ±0.00 | 0.81 ±0.00 | 0.60 ±0.00 | - | 0.49 ±0.00 |
| | DP-SGD | 0.52 ±0.07 | 0.61 ±0.05 | 0.33 ±0.05 | - | 0.29 ±0.34 |
| | PATE | 0.47 ±0.08 | 0.49 ±0.03 | 0.26 ±0.02 | - | 0.51 ±0.19 |
| | MERF | **0.95** ±0.02 | 0.65 ±0.04 | 0.32 ±0.03 | - | **0.65** ±0.05 |
| TravelTime | Algorithm 1 | 0.49 ±0.01 | **0.75** ±0.02 | **0.45** ±0.02 | 0.78 ±0.03 | 0.39 ±0.15 |
| | SliceWass | 0.30 ±0.02 | 0.62 ±0.02 | 0.33 ±0.01 | **0.93** ±0.01 | 0.46 ±0.17 |
| | DP-SGD | 0.41 ±0.06 | 0.54 ±0.01 | 0.22 ±0.01 | 0.82 ±0.12 | 0.11 ±0.21 |
| | PATE | 0.42 ±0.10 | 0.39 ±0.03 | 0.14 ±0.02 | 0.88 ±0.04 | **0.51** ±0.16 |
| | MERF | **0.56** ±0.08 | 0.37 ±0.03 | 0.08 ±0.01 | 0.62 ±0.01 | 0.11 ±0.18 |

Table 3: We repeat the experiment in Table 1 with a different privacy budget $\epsilon = 8.1$.

