# OpenReview forum: "Privacy without Noisy Gradients: Slicing Mechanism for Generative Model Training"
_NeurIPS.cc/2024/Conference — NeurIPS 2024 poster_

### Official Review · Reviewer_jo82 · 2024-07-11

**Soundness:** 3
**Presentation:** 3
**Contribution:** 3
**Rating:** 6
**Confidence:** 3

**Summary:**

The manuscript introduces a slicing privacy mechanism for training generative models without noisy gradients. This mechanism injects noise into random low-dimensional projections of private data, providing strong differential privacy guarantees. The study introduces the smoothed-sliced
$f$-divergence and a kernel-based estimator for it, allowing the training of generative models without adversarial training. Empirical results show improved synthetic data quality compared to existing methods.

**Strengths:**

- The introduction of the slicing privacy mechanism and smoothed-sliced $f$-divergence represents a novel approach to training privacy-preserving generative models, addressing the limitations of noisy gradient-based methods.
- The paper provides strong theoretical foundations for the proposed methods, including privacy guarantees and statistical consistency of the smoothed-sliced $f$-divergence.
- Extensive experiments demonstrate the method's effectiveness in generating high-quality synthetic data across various datasets, outperforming baseline methods.

**Weaknesses:**

- This is a very solid piece of work. The proposed method is simple yet effective, and I have no particular concerns or issues with it.

**Questions:**

None

**Limitations:**

Yes

---

> ### Author Rebuttal · Authors · 2024-08-06
>
> We thank the reviewer for the thoughtful review and for appreciating the merits of the work! If there are any further questions that might lead to improving your score please let us know.

---

### Official Review · Reviewer_JoQ4 · 2024-07-12

**Soundness:** 3
**Presentation:** 2
**Contribution:** 3
**Rating:** 4
**Confidence:** 3

**Summary:**

The paper proposes to add noise to randomly projected private data along with optimizing a newly proposed metric smoothed-sliced f-divergence to train generative models. Such paradigm can circumvent adding noise to gradient and enable more architecture choices. Experiment show the proposed method perform competitively with existing methods.

**Strengths:**

1. The proposed approach directly add noise to data, thus enable more data processing possibilities on privatized data.
2. The proposed approach is shown to have consistency and perform competitively with baseline algorithms such as DP-SGD, PATE, and SliceWass.

**Weaknesses:**

1.I feel thee baseline algorithms are kind of weak for the setting considered in the paper (synthetic tabular data generation). I am not an expert in this specific area but SliceWass is proposed 3 years ago, DP-SGD and PATE are classical DP algorithms which have many variants improving upon them in recent years,  e.g., Adam also has DP versions.

2. The algorithm is proposed for generative model training, at least it is what claimed in the title and abstract, but it is only tested on tabular data.

**Questions:**

1. Have the authors tested the algorithm on other type of data, such as image or text?
2. Do the authors have explanations why the proposed approach could be better than the baseline algorithms?

**Limitations:**

I did not find any limitation discussion specifically for the proposed approach. Adding it would be beneficial.

---

> ### Author Rebuttal · Authors · 2024-08-07
>
> We appreciate the reviewer’s careful reading of our paper and thoughtful comments!
> ---
> **Q1. Weak baselines**
>
> A1. We have added experimental comparisons to the state-of-the-art MERF method in the attached pdf. In Figure 1 of the attached PDF, we have considered private image generative modeling for MNIST, showing that our approach outperforms the state-of-the-art baseline MERF method for larger privacy budgets. In this experiment, note that MERF, as implemented by the authors thereof, only uses the mean embedding of the real data in each class. This allows them to use much lower noise and achieve better results for low epsilon levels, but damages performance for higher privacy budgets since their approach cannot learn the diversity of each class’s distribution (due to taking the mean only). This explains the trends we see and emphasizes the advantages of our approach that requires no such limiting preprocessing step and can retain the full diversity of the data distribution.
>
> In Table 1 of the PDF, we also add MERF as a baseline to the tabular data experiment in the main text, showing that our proposed method (often significantly) outperforms MERF in 20 out of the 24 metrics across five datasets, including all 10 categorical metrics, 4 out of 5 classifier F1 scores, and 6 ouf of 9 numerical metrics. We used the implementation at https://github.com/ParkLabML/DP-MERF for this experiment.
>
>
> ---
> **Q2.  Other types of data.**
>
> A.2 See the response to the question above, where we expand our treatment of image data (in addition to the domain-adaptation experiments in the original submission for image data).
>
> ---
> **Q3. Why the proposed approach could be better than the baseline algorithms**
>
> A3.  Our intuition is that training generative models can be very unstable and requires careful design of the model architecture. Traditional approaches like DP-SGD, PATE, and their variants ensure differential privacy (DP) by modifying the training algorithm. This modification makes the model hard to converge due to gradient clipping and the noise added to the gradient, leading to potential issues like mode collapse, especially since hyperparameter tuning is challenging under DP constraints. In contrast, our algorithm separates DP from the generative model training by adding noise to low-dimensional projections of real data, which are then used to train the generative models. Although the noisy data may still affect the quality of the synthetic data, hyperparameter tuning becomes much easier, and any optimizer (e.g., SGD) can be applied effectively to optimize the generative model.
>
> ---
> **Q4.  Discussions about limitations.**
>
> A.4. Thank you for highlighting this matter. In response to your concerns, we provide a discussion on the limitations and potential future directions below. We will ensure this discussion is incorporated into the revised paper.
>
> In this paper, we introduce a generic framework for training generative models with differential privacy guarantees. We demonstrate the efficacy of our method through numerical experiments on both tabular and image data. It would be interesting to extend our framework to other types of data, such as natural language processing and time-series data. We establish an asymptotic statistical consistency guarantee for our algorithm. An interesting direction for future research would be to derive finite sample guarantees, specifically sample complexity bounds for our method. Additionally, we believe it is crucial to investigate the capabilities and limitations of private synthetic data from various perspectives. For instance, if a machine learning model trained on synthetic data exhibits algorithmic bias when deployed on real data, it is important to identify the source of this bias and correct the model.
>
>
> We hope the above answers address your concerns, and if any questions remain, please let us know!

---

### Official Review · Reviewer_NMCZ · 2024-07-13

**Soundness:** 3
**Presentation:** 3
**Contribution:** 2
**Rating:** 5
**Confidence:** 3

**Summary:**

This paper proposes a DP generative modeling technique via f-divergence and random projection. Specifically, both real data and synthetic data are randomly projected into a lower-dimensional space, where the noise is added to the aggregation of the projected data such that the effect of individual data point is bounded. Minimizing the smoothed-slicing f-divergence between the noisy embeddings will guarantee the match of two distributions, thereby can be used to train a generative model. Compared to adding noise to gradients (e.g. DP-SGD), this method is more convenient and scalable. Experiments show that it generally outperforms prior related methods in synthesizing tabular data.

**Strengths:**

1. Looking for better alternatives to DP-SGD is an active research area to circumvent the common issues DP-SGD has in practice, thus is well-motivated and of wide interest in the related field.
2. The method is complete, with full description and theoretical privacy analysis.
3. Empirically, multi-dimensional evaluation metrics are considered, and the proposed method outperforms other alternatives in most metrics.

**Weaknesses:**

1. The proposed method is similar to the MMD-based methods (references on line 96) in the following way:
    + MMD-based methods apply MMD as the divergence measure, since MMD(P, Q)=0 iff P=Q, similar to $D_f$(P, Q)=0 iff P=Q, thus both divergence measure can be used to train a generative model.
    + MMD-based methods minimize the distance between kernel mean embeddings (KME) of real data and generated data, while noise is added to the KME to retain DP guarantee. In this work, both real and generated data are projected to an embedding space, and noise is injected into the embeddings to retain DP guarantee, so the idea of the overall paradigm is quite similar.
    + KME in theory is an infinite-dimensional embedding, which is not convenient to add Gaussian noise, so those works propose to approximate KME by projecting it into a finite-dimensional space. In this work, in theory you have infinitely many directions to slice where only a finite number ($m$) of directions are randomly chosen. So both of you apply the finite-projection idea and therefore both of you will have approximation errors.

In light of the above similarity, the current discussion on the difference and association with MMD-based methods are not adequate. Also, it is more ideal to include some of those methods in the empirical comparison (Table 1), because they ran experiments on tabular dataset as well.

2. For a work in DP, I did not even find sensitivity analysis, which is crucial for proving DP guarantees. Even if it is trivial it should be included for completeness.

3. It looks to me that adding noise to synthetic data (eq. 2) is unnecessary, and will hurt the model utility. Can you please explain why?

**Questions:**

1. Existing methods move to adversarial dual form due to "...they often suffer from scalability issues and are not friendly to gradient-based optimization..." Now you circumvent the adversarial training, then does the proposed method suffer from scalability issues or not friendly to gradient-based optimization?
2. Maybe I am out of the field, but what does "slicing and smoothing" (line 116) exactly mean? What are the differences? And you term the proposed divergence "smoothed-sliced f-divergence", what does "smoothed-sliced" mean?
3. In remark 1 the authors claim that [RL21] did something wrong in their derivation because the slicing matrix U is not included in the privacy mechanism. I am not familiar with [RL21], but is U independent of the dataset X? Why will U affect the privacy analysis?
4. Line 221, "...we can achieve a tighter privacy bound by reducing a factor of..." How this factor is derived?
4. The proposed method seems quite universal, i.e. should be scalable to other domains like image datasets as well, is it true?

---

> ### Author Rebuttal · Authors · 2024-08-07
>
> PART 1
>
> We thank the reviewer for the thoughtful comments and for appreciating the novelty of the work!
>
> ---
> **Q1. Comparison with MMD-based methods.**
>
> A1. Indeed, this is a great point! As mentioned in the general response, we have added new experiments to address this.
>
> In Figure 1 of the attached PDF, we have considered private image generative modeling for MNIST, showing that our approach outperforms the state-of-the-art baseline MERF method for larger privacy budgets (MERF is an MMD-based method). In this experiment, note that MERF, as implemented by the authors thereof, only uses the mean embedding of the real data in each class. This allows them to use much lower noise and achieve better results for low epsilon levels, but damages performance for higher privacy budgets since their approach cannot learn the diversity of each class’s distribution (due to taking the mean only). This explains the trends we see and emphasizes the advantages of our approach that requires no such limiting preprocessing step and can retain the full diversity of the data distribution.
>
> In Table 1 of the PDF, we also add MERF as a baseline to the tabular data experiment in the main text, showing that our proposed method (often significantly) outperforms MERF in 20 out of the 24 metrics across five datasets, including all 10 categorical metrics, 4 out of 5 classifier F1 scores, and 6 ouf of 9 numerical metrics. We used the implementation at https://github.com/ParkLabML/DP-MERF for this experiment.
>
>
> ---
> **KME in theory is an infinite-dimensional embedding, which is not convenient to add Gaussian noise, so those works propose to approximate KME by projecting it into a finite-dimensional space. In this work, in theory you have infinitely many directions to slice where only a finite number (m) of directions are randomly chosen. So both of you apply the finite-projection idea and therefore both of you will have approximation errors.**
>
> Note that using a finite number of slices is a very different form of approximation than simply projecting onto a finite-dimensional space. As a result, our approach is much more amenable to theoretical guarantees. Firstly, in our setting, any errors from finite numbers of slices can be easily controlled since the slices are independent and identically distributed, while errors from truncating an infinite-dimensional space to a few dimensions is difficult to control theoretically. Secondly, note that our total projection dimension (number of slices X slice dimension) is actually fairly large, oftentimes on the order or larger than the ambient dimension of the data itself. This should imply that the loss of distributional information due to slicing is quite minimal indeed.
>
> ---
> **Q2.  Sensitivity analysis**
>
> A.2. You are absolutely right. In most DP work, particularly when noise is added to a statistical query, a sensitivity analysis lemma is derived to prove the DP theorem. However, our privacy mechanism differs as it is not an additive mechanism; instead, it uses Gaussian noise to randomly project data into a lower-dimensional space and then adding noise. We established our (Renyi)-DP guarantee by directly bounding the Renyi divergence (Lemma 2 in Appendix), and leveraging a prior result that bounds traditional epsilon-delta differential privacy in terms of Renyi-DP. This strategy significantly simplifies our proof. Similar ways for proving DP theorem (without a sensitivity lemma) have been used in other privacy mechanisms, such as in the DP proof for the Johnson-Lindenstrauss transform in [BBDS12].
>
> ---
> **Q3. Adding noise to synthetic data (eq. 2) is unnecessary and will hurt the model utility**
>
> A3. Adding noise to synthetic data may seem counterintuitive, but it is essential for our approach to ensure that the learned model is consistent with the true data distribution. Recall that we minimize the loss function, specifically the smoothed-slicing f-divergence (SD), between the distributions of synthetic and real data. Proposition 1 states that SD = 0 iff the real and synthetic data distributions perfectly match. This property holds only if we apply the same amount of noise to both the real and synthetic data distributions.
>
> To illustrate this intuition, consider an example: let the real data follow a 1D normal distribution $N(0,1)$ and the synthetic data follow $N(0, \sigma)$ with a trainable parameter $\sigma$. If we add noise only to the real data distribution, for instance, Gaussian noise $N(0, \sigma_{\text{noise}})$, minimizing our loss function to zero will lead to the synthetic data having a greater variance than the real data: $\sigma = 1 + \sigma_{\text{noise}} > 1$.
>
> ---
> **Q4.  Does the proposed method suffer from scalability issues or not friendly to gradient-based optimization?**
>
> A.4. Our proposed method should indeed not suffer from these issues, as we circumvent the need for noisy gradients and our objective function is well-behaved and not a min-max problem.
>
> ---
> **Q5. What does "slicing and smoothing" (line 116) exactly mean?**
>
> A5. “Slicing” refers to (randomly) projecting high-dimensional data into lower-dimensional spaces, like slicing a loaf of bread into thinner pieces. “Smoothing” refers to adding Gaussian noise, which makes the resulting distribution less peaked and more spread out.
>
> We term the proposed divergence “smoothed-sliced f-divergence” since it (randomly) projects the original and synthetic data  distributions onto lower-dimensional spaces (i.e., slicing), followed by adding isotropic Gaussian noise (i.e., smoothing), and averaging their f-divergence over all projections.

---

> > ### Comment · Reviewer_NMCZ · 2024-08-12
> >
> > Thanks for the reply. Discussion on Q1 needs to be included in the revision. I am generally satisfied with the responses except Q2.
> >
> > I disagree with the answer to Q2. I don't get why eq(3) is not an additive mechanism. For the papers you mentioned, both [BBDS12] and [EKKL20] made assumptions on the sensitivity of the output, i.e. <= 1, whereas I don't find similar assumptions in this paper. In fact, the random slicing UX is unbounded. It looks to me that you need to clip the output and then add V that is scaled with the sensitivity.

---

> > > ### Author Response · Authors · 2024-08-12
> > > **Response and clarification on bounded norm**
> > >
> > > Thank you for your prompt response! We are glad to hear that you are generally satisfied with our responses. We will ensure that the discussion on Q1 and the experimental results are included in our revised paper.
> > >
> > > Regarding Q2, you are absolutely right that UX would be unbounded without additional assumptions and privacy guarantees would not hold, we apologize for misunderstanding the precise point of your previous statement of Q2. To clarify---our theorem explicitly requires each record in X to have a norm <= 1 (please refer to Line 145 in our paper, and Lines 659–664 for a discussion on how we satisfy this assumption in our experiments). Given this, UX will be bounded (with high probability) since U is a Gaussian matrix with a specified covariance matrix. Hence the sensitivity is immediately controlled allowing us to compute the correct variance of the additive Gaussian noise applied after slicing to achieve the desired privacy guarantees. While our proof strategy goes through Renyi divergence (crucially using the norm bound assumed for X) rather than through a more explicit sensitivity analysis for the linear U mechanism, the principle is fundamentally the same.
> > >
> > > Similarly, as you pointed out, the privacy analyses in [BBDS12, EKKL20] rely on the same bounded norm assumption. In the revision, we will add a comment pointing out that this norm bound is key and analogous to sensitivity discussions in other works.

---

> > > > ### Comment · Reviewer_NMCZ · 2024-08-13
> > > >
> > > > Thanks for the explanation. I have increased my score

---

> > > > > ### Author Response · Authors · 2024-08-13
> > > > > **Thanks!**
> > > > >
> > > > > Thanks for increasing your score! We note, however, that if we have indeed addressed all your concerns, a score of 5 is is still tending towards rejection in reality, and - according to the review guidelines - should be used sparingly. If there are any further concerns you have that we could address, please let us know! Otherwise, we respectfully ask that your score reflect your positive evaluation.
> > > > >
> > > > > Thanks!
> > > > > the authors

---

> ### Author Response · Authors · 2024-08-07
> **Part 2**
>
> ---
> **Q6.  In remark 1 the authors claim that [RL21] did something wrong in their derivation because the slicing matrix U is not included in the privacy mechanism. Is U independent of the dataset X? Why will U affect the privacy analysis?**
>
> A.6. Yes, U is generated independently of the dataset X, but the sliced output UX is clearly dependent on U, hence conditioned on (UX), U and X are now dependent! Since the generative model training requires both the projected data and the projection directions (i.e., the U matrix must be known to the model during training), not accounting for the fact U is known can result in privacy leakage.
> As an analogy, let’s consider the simple additive noise setting, where the output from a Laplace mechanism is used. In such a case, any downstream operation must not access the Laplace noise added by the privacy mechanism. If it did, the noise could be used to denoise the answer, even if the noise is independent of the real data. This is a crucial oversight, which we remedy by explicitly including all needed factors in the privacy analysis.
>
> ---
> **Q7. Line 221, "...we can achieve a tighter privacy bound by reducing a factor of..." How this factor is derived?**
>
> A7. The intuition is that a deterministic U corresponds to a scenario where the adversary can engage with the design of the privacy mechanism and choose how to project the data. This grants the adversary more freedom to influence the design of the privacy mechanism, resulting in a less effective privacy protection.
>
> Technically, we derive this factor by comparing the Renyi divergence between two privacy mechanisms, where one has random U and the other has deterministic U (see Proposition 3 in the appendix).
>
> ---
> **Q8.  Is the proposed approach scalable to other domains like image datasets as well?**
>
> A.8. Thank you for bringing up this concern. Indeed, our proposed approach is applicable and scalable to other domains, including image data. For instance, please refer to our domain adaptation experiment (Section 4.2), where we apply our method to MNIST and USPS datasets.
>
> In Figure 1 of the attached PDF, we also have considered private image generative modeling for MNIST, showing that our approach outperforms the state-of-the-art baseline MERF method for larger privacy budgets. In this experiment, note that MERF, as implemented by the authors thereof, only uses the mean embedding of the real data in each class. This allows them to use much lower noise and achieve better results for low epsilon levels, but damages performance for higher privacy budgets since their approach cannot learn the diversity of each class’s distribution (due to taking the mean only). This explains the trends we see and emphasizes the advantages of our approach that requires no such limiting preprocessing step and can retain the full diversity of the data distribution.
>
> In Table 1 of the PDF, we add MERF to the experiment in the main text, showing that our proposed method (often significantly) outperforms MERF in 20 out of the 24 metrics across five datasets, including all 10 categorical metrics, 4 out of 5 classifier F1 scores, and 6 ouf of 9 numerical metrics. We used the implementation at https://github.com/ParkLabML/DP-MERF for this experiment.
>
> We hope the above answers clarify the reviewer’s view of our work, and if any questions remain, please let us know!

---

### Author Rebuttal · Authors · 2024-08-07

General response:

We thank the reviewers for their thoughtful and helpful comments. Due to the suggestions of several reviewers, in the attached pdf we have included results for two new experiments showing the advantage of our methods over the state-of-the-art MMD-based MERF method. We will add both of these experiments to the revision.

Firstly, in Figure 1 of the attached PDF, we also have considered private image generative modeling for MNIST, showing that our approach outperforms the state-of-the-art baseline MERF method for larger privacy budgets. In this experiment, note that MERF, as implemented by the authors thereof, only uses the mean embedding of the real data in each class. This allows them to use much lower noise and achieve better results for low epsilon levels, but damages performance for higher privacy budgets since their approach cannot learn the diversity of each class’s distribution (due to taking the mean only). This explains the trends we see and emphasizes the advantages of our approach that requires no such limiting preprocessing step and can retain the full diversity of the data distribution.

In Table 1 of the PDF, we add MERF as a baseline to the tabular data experiment in the main text, showing that our proposed method (often significantly) outperforms MERF in 20 out of the 24 metrics across five datasets, including all 10 categorical metrics, 4 out of 5 classifier F1 scores, and 6 ouf of 9 numerical metrics. We used the implementation at https://github.com/ParkLabML/DP-MERF for this experiment.

---

### Author Response · Authors · 2024-08-12
**Responses to the rebuttal**

We thank all Reviewers for their time and thoughtful feedback! We are glad to see our paper was positively received, and the Reviewers found that: the paper is novel, rigorous, and well-written (all Reviewers); it is well-motivated and of wide interest in the related field (Reviewer NMCZ); the proposed approach is shown to have consistency and perform competitively with baseline (Reviewer JoQ4); the paper provides strong theoretical foundations (Reviewer jo82). We also recognize that the Reviewers are busy handling multiple papers, so their thoughtful feedback is even more appreciated.

In response to the Reviewers’ questions, we did our best to provide comprehensive answers and included additional experimental results (please refer to the 1-page pdf in the global response). Please do follow up with us if you have additional suggestions and feedback that can further strengthen the paper. Thanks!

---

### Decision · Program_Chairs · 2024-09-25

**Decision:**

Accept (poster)

**Comment:**

The paper is on the borderline. The discussion turned out in the authors' favor and I am happy to recommend "accept".

The two main criticisms from the reviewers are:
1. (from JoQ4) Weak experimental evaluation. Why generative model for tabular data only?
2. (from NMCZ) Unclear sensitivity analysis of the differentially private mechanism.

The authors' rebuttal were mostly satisfactory in my opinion. Generative models for tabular data are still actively being researched on when we want to generate synthetic data differentially privately.  Of course, it is not wrong that the title is a bit click-baiting and the content may disappoint readers hoping to read about large language models.  It's not the authors' fault, but a disclaimer early on could be useful.

As for the second criticism, it was resolved during AC-Reviewer discussion.  The proposed mechanism and its RDP analysis appear to be correct. It is not a purely noise-adding mechanism, thus does not have the notion of a "query" nor does it require a sensitivity analysis.

Finally, I think the main result of the work is interesting (though it is somewhat hidden in the bells and whistles of "multiple-slices" and "generative models"). I think it is worth emphasizing the following result more crisply (which I summarize in English)

> Random projection matrix results in (sometimes) tighter privacy guarantees than the alternative simple analysis that conditions on the projection matrix.

> The key observation is that a random projection matrix symmetrizes the distribution so that for both X and X' the mean is 0.

I do have a related question:

- If you orthogonalize the iid gaussian matrix, the above main result should still work? Because at the limit of d --> inf,  Gaussian random matrix with your scaling converges to an orthonormal matrix. It makes the analysis harder, but if it works out, then the result is a lot stronger because for large \alpha, the RDP is still finite.